# Association between community-based resource collection site use and functional disability risk among older adults: A Quasi-experimental study

Noriyuki Abe[1]*, Kazushige Ide[2], Kenjiro Kawaguchi[1], Daisuke Kumazawa[3], Katsunori Kondo[2,4,5]

1 Department of Social Preventive Medical Sciences, Center for Preventive Medical Sciences, Chiba University, Chiba, Japan, 2 Department of Community Building for Well-being, Center for Preventive Medical Sciences, Chiba University, Chiba, Japan, 3 Center for Innovation Policy, Kanagawa University of Human Services, Kanagawa, Japan, 4 Research Department, Institute for Health Economics and Policy, Tokyo, Japan, 5 Department of Social Impact Assessment and Evaluation, Graduate School of Medicine, Kyoto University, Kyoto, Japan

* abe.n@chiba-u.jp

## Abstract

This study aimed to investigate the longitudinal association between users versus non-users of MEGURU STATION, a community-based resource collection site, and the risk of functional disability among older adults in Japan. This quasi-experimental study included 973 older adults aged ≥65 years from three communities in Japan. Baseline and follow-up surveys were conducted 1 year apart to measure the risk scores for functional disability (RSFD) as the primary outcome. The main explanatory measure was self-reported MEGURU STATION use, with participants categorized as users or non-users. Mixed-effects linear regression models accounted for community-level variability and were adjusted for covariates, including sex, age, activities of daily living (ADL), education, subjective economic status, residential status, employment, and social participation. An additional analysis examined changes in going out, social interaction, and participation in community activities associated with MEGURU STATION use. Of the participants, 19.2% reported MEGURU STATION use. MEGURU STATION use was associated with a lower RSFD (B = −1.20, 95% confidence interval: −2.27, −0.12). Users reported increased opportunities for social interaction, participation in community activities, and going out compared with non-users. In summary, MEGURU STATION, a community-based intervention that integrates social interaction into daily routines, lowers the risk of functional disability among older adults. This scalable and socially inclusive model holds promise for promoting healthy aging. Future research should investigate its long-term impact and cultural adaptability.

**Data availability statement:** The dataset used in this study cannot be publicly shared due to ethical and legal restrictions. Although participant consent was obtained for research use, it did not include permission for public data sharing. Furthermore, the data are owned by local insurers and were obtained under joint research agreements. Access to the data is restricted by the personal information protection committees of the respective municipalities. Qualified researchers may request access to the anonymized dataset by contacting the Research Ethics Committee of Chiba University (email: inohana-rinri@chiba-u.jp). Additionally, the questionnaire used in this study was developed based on the JAGES framework (https://www.jages.net/), but permission to publish it has not been granted by the relevant municipality.

**Funding:** 1) Open Innovation Platform with Enterprises, Research Institute and Academia [OPERA, grant number JPMJOP1831] of Japan Science and Technology (JST). 2) Amita Holdings Co., Ltd. The funders had no role in study design, data collection and analysis, decision to publish, or preparation of the manuscript.

**Competing interests:** The authors have declared that no competing interests exist.

## Introduction

The global population is aging rapidly, with the proportion of individuals aged over 60 years projected to nearly double from 12% to 22% between 2015 and 2050 [1]. Japan has the highest aging rate among all countries. Consequently, developing effective strategies to prevent functional disability and achieve healthy aging has become an important public health priority.

Social participation is crucial for maintaining and improving the health of older adults, especially in preventing functional disabilities. Social participation is defined as engagement in activities that involve interacting with others in social or community settings [2]. The "Age-friendly Cities (AFC)" initiative, proposed by the World Health Organization, identifies social participation as one of eight essential domains [3]. Extensive evidence suggests that social participation can help protect against declining physical and mental health, such as incident disability, functional and cognitive decline [4–8], and mortality [9,10]. Furthermore, social participation was associated with a lower risk of incident disability and mortality in both healthy and frail older adults. Older adults with frailty are at a higher risk of functional disability and mortality [11]. Face-to-face interactions have been shown to reduce feelings of loneliness and depression among older adults [12]. Previous studies have highlighted that social participation is more common among socially active older adults. In contrast, socially inactive individuals face greater barriers to participation, making strategies to encourage their involvement a critical issue [13].

Primordial prevention refers to interventions aimed at addressing the root causes of health issues, including socioeconomic, cultural, political, and environmental factors [14]. Therefore, creating an environment that fosters social participation without relying on older adults' health consciousness is crucial. Previous research has demonstrated that environmental factors, such as proximity to services, transportation, and a higher concentration of older adults, are strongly associated with increased social participation [15]. One practical application of primordial prevention is the establishment of community gathering places that serve as central hubs where older adults living in the area gather for activities such as exercise, hobbies, and other social engagements [16]. These initiatives, supported by local governments in Japan, aim to promote social participation, build social capital, and prevent the onset of functional disabilities among older adults [16,17]. Previous studies demonstrated that community gatherings promote social participation [16] and prevent the onset of functional disabilities and dementia [4,5]. These findings suggest that providing a comfortable environment for social interaction among older adults may be an effective strategy for increasing social participation and preventing the onset of functional disabilities. However, in Japan, few men and high-frequency users utilize these facilities, and developing more diverse venues has been identified as a challenge [18].

Using a resource collection site to promote social participation exemplifies the creation of an environment that fosters social interaction through primordial prevention. In Japan, where waste segregation is highly developed, residents are required to separate recyclable and non-recyclable materials and transfer them to nearby waste collection points. "MEGURU STATION" is a local community hub centered on

resource collection, where household waste is sorted and collected for recycling [19]. MEGURU STATION has established multiple community functions beyond waste management. Residents can visit the hub to sort recyclable waste, purchase fresh vegetables at vegetable stands, converse with neighbors, or relax on benches [19]. Through these diverse activities, MEGURU STATION is expected to foster social participation and potentially reduce the risk of functional disability among users. This approach aligns with the concepts of primordial prevention and social participation. We hypothesized that the use of MEGURU STATION can reduce the risk of functional disability. This study aimed to investigate the longitudinal association between MEGURU STATION use versus non-use and the risk of functional disability among older adults in Japan.

## Materials and methods

### Ethical approval

All procedures were performed in compliance with relevant laws and institutional guidelines and were approved by the relevant ethics committees at Chiba University (approval numbers: 3442 and M10260). The privacy rights of human participants have been observed. The participants were informed that checking the acceptance checkbox and returning the questionnaire would signify their consent to participate.

### Study design

This study employed a quasi-experimental design to evaluate the effects of MEGURU STATION use on the risk of functional disability among older adults. The study utilized data from a self-administered mail survey conducted at two time points: before (baseline) and 1 year after (follow-up) MEGURU STATION installation. The intervention was implemented in the following communities: Community A (anonymized at the request of Ikoma City) in Ikoma City, Nara Prefecture and the Hongo and Ozeki communities in Tachiarai Town, Mitsui County, Fukuoka Prefecture. In 2020, the proportion of people aged ≥65 years was 28.7% in Ikoma City and 28.0% in Tachiarai Town [20].

### Study setting and participants

This study targeted older adults aged ≥65 years who resided in the three communities where the MEGURU STATION intervention was implemented. The eligibility criteria included individuals not officially certified as requiring long-term care insurance (LTCI) services [21] in Japan. The exclusion criteria included individuals who were unable to provide consent to participate in the study, those whose self-reported sex and age did not match the information in the registry held by the local government, those aged <65 years, and those already certified as requiring LTCI.

A self-administered questionnaire was mailed and hand-delivered to all eligible residents (n = 2,554) in the three communities. A complete enumeration survey was conducted in Community A and the Ozeki community. In the Hongo community, both hand delivery of surveys to some MEGURU STATION users and a complete enumeration survey were conducted. Baseline surveys were conducted in November 2020 in Community A, May 2022 in the Hongo community, and September 2022 in the Ozeki community. MEGURU STATION was installed in December 2020 in Community A, April 2022 in the Hongo community, and September 2022 in the Ozeki community. Follow-up surveys were conducted approximately 1 year after installation: November 2021 in Community A, July 2023 in the Hongo community, and December 2023 in the Ozeki community. Of 1,524 baseline survey respondents (response rate, 59.7%), 80 individuals were excluded: 13 due to discrepancies between their self-reported sex and age and the information in the registry held by the local government, 8 for being aged <65 years, 11 for being certified as requiring support or nursing care, and 48 for not providing consent to participate in the study. Of 1,443 individuals who met the eligibility criteria, 993 (follow-up rate, 68.8%) responded. Non-consenters (n = 20) in the follow-up study were excluded, and 973 participants were included in the final analysis (Fig 1).

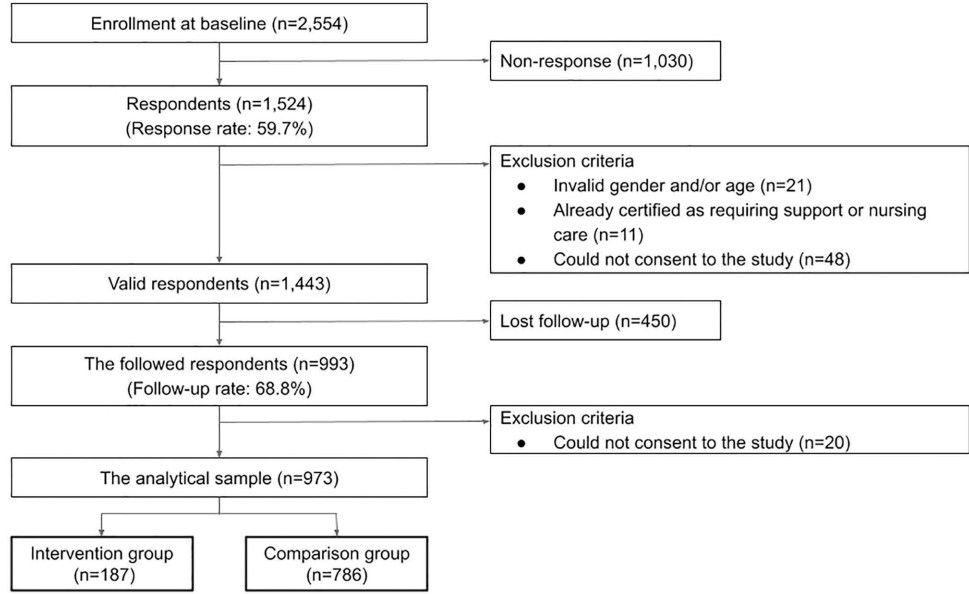

**Fig 1. Flowchart of participant selection and exclusion criteria.**

## Intervention

MEGURU STATION was established as a community hub aimed at using daily activities, such as garbage disposal, to foster mutually supportive communities and prevent functional disability among older adults [19].

All MEGURU STATION locations were near community centers, serving as community gathering places for residents. While resource collection was the core function, each MEGURU STATION incorporated unique peripheral functions tailored to community needs and policies. Core functions refer to main services directly related to resource collection (e.g., sorting and disposing of recycle waste). For example, Community A features a place where people can relax on benches with coffee (greenway café), thus creating opportunities for social interaction. In the Hongo community, volunteers cultivate crops on a small farm (MEGURU Farm), where the vegetables grown are shared and sold. In the Ozeki community, the residents sell cakes. The key concept is common to all three communities, but the peripheral functions vary according to each community's policies. Details of the peripheral functions and specific configurations of MEGURU STATION in each community are provided in S1 Fig and S1 Table.

Participants were classified as MEGURU STATION users (intervention group) if they reported using MEGURU STATION and peripheral functions during the follow-up period. The classification of MEGURU STATION users was based on self-reported data from the follow-up survey. Participants were asked to select all activities they had engaged in at the community center, from a list including both MEGURU STATION-related functions (e.g., resource collection, vegetable sales, café use, greenway relaxation) and unrelated programs (e.g., community salons, hobby clubs). Only those who selected at least one predefined activity officially designated as part of the MEGURU STATION program by municipal authorities were classified as users. These activities were identified in advance through consultation with the respective local governments and documentation of MEGURU STATION operations.

While usage logs or observational data were not available owing to privacy concerns and the absence of consent for research use, MEGURU STATION facilities were widely promoted through local newsletters and community announcements and were familiar landmarks in participants' daily routines. Therefore, although the possibility of self-report bias cannot be entirely excluded, we believe that participants could reliably recognize and report their use of MEGURU STATION.

## Comparison group

The comparison group comprised participants who did not use MEGURU STATION during the follow-up period. Participants were classified as non-users if they indicated in the follow-up survey that they did not use MEGURU STATION.

## Outcomes

The primary outcome was the risk of functional disability, as assessed using risk scores for functional disability (RSFD) [22,23]. The RSFD is a validated tool for predicting the likelihood of future LTCI certification. It comprises 12 items, including instrumental activities of daily living (IADL), motor function, nutritional status, frequency of outings, sex, and age. The RSFD score ranges from 0 to 48, with higher scores indicating a greater risk of functional disability [22]. The predictive validity of the RSFD was established using longitudinal data collected over 3 years (2013–2016) from 23 municipalities of varying sizes in Japan, showing a sensitivity of 0.73 and a specificity of 0.73 for LTCI certification [22].

## Covariates

Ten covariates were employed based on previous studies on social participation and functional disability [6,7,16]: baseline sex; age; activities of daily living (ADL); education; subjective economic status; residential status; employment; social participation (at least once a month in one or more of the following activities—sports, volunteer, community gathering place, hobbies, learning or cultural group, senior citizen clubs, and neighborhood association); and the RSFD. The RSFD was included as a covariate to control for participants' initial differences in functional status and to account for potential regression to the mean.

## Statistical analysis

First, baseline descriptive statistics were calculated to compare characteristics between MEGURU STATION users and non-users, as well as to assess the utilization status of the MEGURU STATION by the community. Next, a mixed-effects linear regression model was used to examine the association between resource collection site use and the RSFD, with communities specified as random effects to account for between-community variations. Regression coefficients (unstandardized beta; B), 95% confidence intervals (CIs), and p-values for RSFD associated with MEGURU STATION use were calculated. We assessed model fit using residual diagnostics.

   Previous studies indicated that participants in community gatherings are more likely to be women or older [16]. Considering these findings, we conducted stratified analyses by sex and age to better understand the potential differences in participation patterns. The missing values for the covariates ranged from 1.8% (sex) to 11.8% (RSFD at follow-up). Because the RSFD is a composite index based on multiple items, a relatively higher rate of missingness was observed. Excluding RSFD, the highest missing rate among covariates was 4.6% for participation in sports groups. To address potential bias due to missing data, we performed multiple imputations using the multivariate normal regression method, assuming that the data were missing at random. We created 20 imputed datasets and combined effect estimates using Rubin's rule [24]. Sensitivity analyses were also performed. We calculated E-values (CI limit) for each exposure-outcome association to evaluate the robustness of the estimated associations with unmeasured confounders [25]. The E-value (CI limit) measures how much unmeasured confounding in an observational study would explain the observed association [25]. Typically, the E-value (CI limit) was calculated based on the risk ratio (RR). In this study, the RR and 95% CI approximations were calculated from the results of linear regression analysis, as previously described [25].

   To examine the mechanisms from intervention to outcome, we assessed whether MEGURU STATION use increased social participation and interaction with others. Participants were asked the following question: 'Since 6 months before the installation of MEGURU STATION, have you noticed any changes in your opportunities?' The distribution of responses was analyzed based on whether participants used MEGURU STATION. The response categories were as follows:

- Opportunities to interact with others: greatly increased, increased, hardly increased, or not increased.

- Age groups of people you interacted with (select all that apply): 12 years or younger, 13–18 years, 19–39 years, 40–64 years, 65–74 years, and/or ≥75 years.

- Opportunities to participate in community activities: greatly increased, increased, hardly increased, or not increased.

- Opportunities to go out: greatly increased, increased, hardly increased, or not increased.

The significance level was set at 5%. All analyses were performed using Stata MP 17 software (StataCorp, College Station, TX, USA).

## Results

The baseline characteristics of the study participants are shown in Table 1. MEGURU STATION users were more likely to be older, female, and independent in ADL compared with non-users. Additionally, users had higher participation rates in all seven types of social activities. One year after the establishment of MEGURU STATION, RSFD increased by 0.1 points for users and 0.7 points for non-users. Table 2 presents the utilization status of MEGURU STATION by each community. In Ikoma (Community A) and Hongo, most users visited MEGURU STATION at least once a week, whereas in Ozeki, usage occurred primarily one to three times per month. In Ikoma, access to MEGURU STATION was predominantly by foot, whereas in Hongo and Ozeki, users arrived primarily by car. Notably, a larger proportion of Ikoma users lived within a 5-minute walking distance of MEGURU STATION, while many Hongo and Ozeki users resided within a 15–30-minute walking distance. In Ikoma, the main purposes of visits included resource collection, check-ins, and organic waste disposal, while Hongo and Ozeki users primarily utilized MEGURU STATION for resource collection.

The association between MEGURU STATION use and subsequent RSFD was analyzed overall (Table 3) and stratified by sex and age (Table 4). Overall, MEGURU STATION use was associated with a lower RSFD (B = −1.20, 95% CI: −2.27, −0.12). In the sex-stratified analysis, no association was observed among men. Among women, MEGURU STATION use was associated with a lower RSFD, although the CI included the null value. In the age-stratified analysis, participants aged 65–74 years also showed a lower RSFD; however, the CI crossed the null. No clear association was observed among participants aged ≥75 years. To assess model fit, we performed residual diagnostics; the results are presented in S2 Fig.

Sensitivity analysis using E-values was used to assess the robustness of the associations with unmeasured confounders. In the overall analysis, the E-value (CI limit) of MEGURU STATION use relative to the RSFD at follow-up was 31.44 (2.01). This means that if unmeasured confounders were associated with a risk ratio of 2.01 times for both the RSFD and MEGURU STATION, the 95% CI would include 1.00. Among the variables entered into the present linear mixed-effects model, only age (51.53) and neighborhood association participation (2.99) had risk ratios above the CI limit of 2.01.

Additional analyses assessed the subjective changes that the participants noticed, such as social participation and interaction with others, among MEGURU STATION users (S2 Table). Participants reported significantly increased opportunities for social interaction (use 43.0% vs non-use 22.7%, p < 0.001), participation in community activities (use 33.7% vs non-use 17.2%, p < 0.001), and going out (use 43.9% vs non-use 27.6%, p < 0.001), particularly in the MEGURU STATION user group. Furthermore, the proportion of MEGURU STATION users participating in sports groups (5.0% points), volunteer groups (3.7% points), community gathering places (3.2% points), and neighborhood associations (4.5% points) increased 1 year after the installation of MEGURU STATION (Table 5).

## Discussion

This is the first study to examine the association between the use of MEGURU STATION, aimed at fostering a mutually supportive community, and the risk of functional disability among older adults. Our findings revealed that MEGURU

**Table 1. Basic characteristics of participants.**

| Variables | Categories | Total | MEGURU STATION | |
|---|---|---|---|---|
| | | | Use | Non-use |
| | | N = 973 | n = 187 | n = 786 |
| Demographic and socioeconomic variables | | | | |
| Age, years | 65–74 | 527 (56.1%) | 124 (69.3%) | 403 (53.0%) |
| Sex | Female | 518 (54.2%) | 115 (63.9%) | 403 (52.0%) |
| Education, years | <10 | 116 (12.3%) | 15 (8.2%) | 101 (13.3%) |
| | 10–12 | 486 (51.6%) | 102 (56.0%) | 384 (50.6%) |
| | ≥13 | 339 (36.0%) | 65 (35.7%) | 274 (36.1%) |
| Employment | Employed | 612 (65.7%) | 122 (67.4%) | 490 (65.3%) |
| Living status | Alone | 126 (13.3%) | 19 (10.6%) | 107 (13.9%) |
| Subjective economic status | Difficult | 164 (17.2%) | 24 (13.2%) | 140 (18.2%) |
| | Normal | 599 (63.0%) | 109 (59.9%) | 490 (63.7%) |
| | Comfortable | 188 (19.8%) | 49 (26.9%) | 139 (18.1%) |
| Health and functional variables | | | | |
| ADL | Independent | 891 (93.8%) | 183 (97.9%) | 708 (92.8%) |
| Risk Assessment Score | Baseline | 14.6 (8.6) | 12.3 (7.8) | 15.1 (8.7) |
| | Follow-up | 15.1 (8.6) | 12.4 (7.4) | 15.8 (8.7) |
| Social participation | | | | |
| No participation | Yes | 195 (20.0%) | 16 (8.6%) | 179 (22.8%) |
| Sports group | Participation | 333 (35.9%) | 93 (51.7%) | 240 (32.1%) |
| Volunteer group | Participation | 322 (34.9%) | 108 (60.7%) | 214 (28.8%) |
| Community gathering place | Participation | 201 (21.5%) | 67 (36.8%) | 134 (17.7%) |
| Hobby group | Participation | 369 (39.8%) | 108 (59.3%) | 261 (35.0%) |
| Learning or cultural group | Participation | 179 (19.2%) | 62 (34.3%) | 117 (15.6%) |
| Senior citizen club | Participation | 269 (28.8%) | 63 (34.8%) | 206 (27.4%) |
| Neighborhood association | Participation | 474 (51.2%) | 113 (63.1%) | 361 (48.3%) |
| MEGURU STATION utilization | | | | |
| Installation area | Ikoma (Community A) | 321 (33.0%) | 59 (31.6%) | 262 (33.3%) |
| | Ozeki | 295 (30.3%) | 38 (20.3%) | 257 (32.7%) |
| | Hongo | 357 (36.7%) | 90 (48.1%) | 267 (34.0%) |
| Use of other MEGURU STATION functions* | Yes | 244 (25.1%) | 187 (100.0%) | 57 (7.3%) |

Data are presented as mean (SD) for continuous variables and n (%) for categorical variables.

*Use of other MEGURU STATION functions refers to peripheral features associated with the MEGURU STATION (e.g., café areas, vegetable stands, community events), not including the core activity of waste collection.

ADL, activities of daily living

STATION users had a lower RSFD than non-users 1 year after installation. Furthermore, MEGURU STATION users experienced significantly increased opportunities for social interaction, participation in community activities, and going out. The proportion of users involved in sports groups, volunteer groups, community gatherings, and neighborhood associations was higher than that of non-users.

Our finding that MEGURU STATION use was associated with lower functional disability risk aligns with previous research demonstrating the protective effects of social participation on functional decline in older adults [4–7,9–11]. The observed increase in social interaction and community participation among MEGURU STATION users suggests that similar mechanisms of social participation may be at play. These findings align with the principles of the AFC framework

**Table 2. MEGURU STATION visit status and purpose, stratified by installation area.**

| Variables | Ikoma (Community A) | Hongo | Ozeki |
|---|---|---|---|
| | n = 114 | n = 91 | n = 39 |
| Age (65–74 years) | 57 (50.0%) | 69 (75.8%) | 23 (59.0%) |
| Sex (female) | 65 (57.0%) | 50 (55.0%) | 25 (64.1%) |
| Visit status | | | |
| Visit frequency | | | |
| Once a week or more | 56 (49.2%) | 52 (57.1%) | 10 (25.6%) |
| 1–3 times a month | 29 (25.4%) | 24 (26.4%) | 20 (51.3%) |
| Few times a year | 29 (25.4%) | 15 (16.5%) | 9 (23.1%) |
| Means of visit | | | |
| On foot | 112 (98.2%) | 9 (9.9%) | 1 (2.6%) |
| Bicycle | 1 (0.9%) | 5 (5.5%) | 1 (2.6%) |
| Motorcycle | 0 | 0 | 0 |
| Car (self-driving) | 1 (0.9%) | 71 (78.0%) | 33 (84.6%) |
| Car (driven by someone else) | 0 | 6 (6.6%) | 4 (10.2%) |
| Walking time to MEGURU STATION | | | |
| Within 5 minutes | 77 (67.5%) | 10 (11.2%) | 2 (5.3%) |
| 5–10 minutes | 36 (31.6%) | 28 (31.5%) | 3 (7.9%) |
| 10–15 minutes | 1 (0.9%) | 20 (22.5%) | 8 (21.1%) |
| 15–30 minutes | 0 | 24 (27.0%) | 23 (60.5%) |
| More than 30 minutes | 0 | 7 (7.8%) | 2 (5.2%) |
| Visit purpose* | | | |
| Use of resource collection site | 59 (51.8%) | 90 (98.9%) | 38 (97.4%) |
| Use of vegetable direct sales | 24 (21.1%) | – | – |
| Volunteer | 19 (16.7%) | – | – |
| Check-in card registration | 42 (36.8%) | – | – |
| Use check-in system | 66 (57.9%) | – | – |
| Bringing kitchen waste | 44 (38.6%) | – | – |
| Free distribution of liquid fertilizer | 26 (22.8%) | – | – |
| Relaxed on the greenway or a nearby bench | 31 (27.2%) | – | – |
| Use of Greenway Café | 23 (20.2%) | – | – |
| Event participation | 18 (15.8%) | – | – |
| Participation in the AOZORA market | – | 3 (3.3%) | – |
| Participation in MEGURU farm | – | 8 (8.8%) | – |
| Participation in cake sales | – | – | 3 (7.7%) |

*Multiple answers were allowed. See S1 Table for details of these functions.

-: No choice exists

[3], which emphasizes social participation as a critical domain. Additionally, our finding that proximity and accessibility to MEGURU STATION can influence use supports existing literature on environmental facilitators of social participation among older adults [26–30]. This highlights how social participation in routine activities can reduce social isolation and foster community engagement. Furthermore, this finding may be interpreted considering the validation study of RSFD. According to Tsuji et al, a one-point increase in RSFD was associated with a 12.8% increase in the risk of developing functional disability over three years, based on a longitudinal cohort study conducted across multiple municipalities in Japan [22]. In this context, the observed reduction of 1.2 points in RSFD among MEGURU STATION users may suggest

**Table 3. Association between the use of MEGURU STATION and RSFD after 1 year.**

| Variables | Categories | B [95% CI] | SE | p-value |
|---|---|---|---|---|
| MEGURU STATION | Non-use | Ref. | | |
| | Use | −1.20 [−2.27, −0.12] | 0.55 | 0.029 |

Note: This model was adjusted for age, sex, years of education, employment, living status, subjective economic status, ADL, RSFD*, social participation (sports group, volunteer group, community gathering place, hobby group, learning and cultural group, senior citizen club, and neighborhood association), and the use of other MEGURU STATION functions.

ADL, activities of daily living; B, unstandardized beta; CI, confidence interval; Ref, reference; RSFD, risk score of functional disability; SE, standard error

*Baseline scores were used.

**Table 4. Stratified analysis between the use of MEGURU STATION and RSFD after 1 year.**

| Variables | Coef. [95% CI] | SE | p-value |
|---|---|---|---|
| Stratified by age | | | |
| 65–74 years | −1.40 [−2.84, 0.04] | 0.74 | 0.057 |
| ≥75 years | −1.31 [−2.95, 0.34] | 0.84 | 0.120 |
| Stratified by sex | | | |
| Male | −0.63 [−2.14, 0.89] | 0.77 | 0.417 |
| Female | −1.54 [−3.09, 0.02] | 0.79 | 0.052 |

Note: All models were adjusted for age, sex, years of education, employment, living status, subjective economic status, ADL, RSFD, social participation (sports group, volunteer group, community gathering place, hobby group, learning or cultural group, senior citizen club, and neighborhood association), and the use of other MEGURU STATION functions. Reference: MEGURU STATION non-user groups within each subgroup.

Coef., coefficient; CI, confidence interval; RSFD, risk score for functional disability; SE, standard error.

**Table 5. Changes in social participation rates owing to MEGURU STATION use.**

| | Total (N = 973) | | MEGURU STATION | | | |
|---|---|---|---|---|---|---|
| | | | Use (n = 187) | | Non-use (n = 786) | |
| | Baseline | Follow-up | Baseline | Follow-up | Baseline | Follow-up |
| Sports group | 333 (35.9%) | 337 (38.2%) | 93 (51.7%) | 97 (56.7%) | 240 (32.1%) | 240 (33.8%) |
| Volunteer group | 322 (34.9%) | 328 (37.2%) | 108 (60.7%) | 114 (63.7%) | 214 (28.8%) | 214 (30.5%) |
| Community gathering place | 201 (21.5%) | 208 (23.3%) | 67 (36.8%) | 70 (40.0%) | 134 (17.7%) | 138 (19.2%) |
| Hobby group | 369 (39.8%) | 359 (40.5%) | 108 (59.3%) | 103 (59.2%) | 261 (35.0%) | 256 (36.0%) |
| Learning or cultural group | 179 (19.2%) | 153 (17.4%) | 62 (34.3%) | 52 (30.2%) | 117 (15.6%) | 101 (14.3%) |
| Senior citizen club | 269 (28.8%) | 258 (29.0%) | 63 (34.8%) | 62 (35.8%) | 206 (27.4%) | 196 (27.3%) |
| Neighborhood association | 474 (51.2%) | 442 (50.2%) | 113 (63.1%) | 117 (67.6%) | 361 (48.3%) | 325 (45.9%) |

Data are presented as n (%)

a potential contribution to lowering the risk of future long-term care certification. However, further research is needed to confirm this association.

Several mechanisms may explain how MEGURU STATION use contributes to lowering the risk of functional disability. First, MEGURU STATION use may increase opportunities to go out and engage in physical activity by adding a community-oriented aspect to routine garbage disposal. Previous studies have clearly shown the association between physical activity and functional disability [31]. This is consistent with our findings that MEGURU STATION users reported more frequent outings than non-users, suggesting that MEGURU STATION contributes to increased outdoor activity. Second, it increases social interactions by offering opportunities to reconnect with acquaintances and make new friends.

MEGURU STATION users reported more opportunities for social interaction than non-users. These social interactions may help mitigate feelings of loneliness and depression, which are known risk factors for functional disability [12,32]. Furthermore, the proportion of MEGURU STATION users participating in sports groups, volunteer groups, community gathering places, and neighborhood associations increased. This suggests that MEGURU STATION use may have contributed to an increase in other forms of social participation and overall lifestyle activity. Third, the cognitive demands involved in sorting recyclable materials and the increased opportunities for conversation may provide cognitive stimulation. While sorting plastics, paper, cans, and bottles requires attention, memory, and executive function, direct evidence linking this activity to cognitive benefit is limited. However, previous studies have shown that activities that incorporate cognitive stimulation can improve cognitive function, social well-being, and mental health [33]. Daily garbage sorting may offer similar benefits and lower the risk of functional disability.

Environmental interventions involve modifying physical and social environments to encourage community interactions and engagement among residents [34,35]. Conventional approaches focus on "hard" infrastructure development, such as creating accessible public spaces [30], parks [27], walking paths [29], greenways [26,28], and community hubs [19]. In contrast, MEGURU STATION represents a unique intervention that combines "hard" elements, such as the installation of the station itself, with "soft" components that transform the routine activity of waste disposal into an opportunity for meaningful social connection within the community. By enhancing waste collection points with social spaces and community amenities, this intervention fostered social participation without requiring significant urban redevelopment. This approach is particularly valuable for older adults, as it addresses mobility limitations while enabling regular opportunities for social interaction. It aligns with previous studies on environmental interventions that have shown increased physical activity, improved social interaction [26,28], and reduced depression [28]. Transforming resource collection sites into spaces for social interaction may serve as a promising environmental intervention. Furthermore, MEGURU STATION could function as a model for a more diverse community gathering place, attracting a larger number of men and older adults aged 65–74 years than traditional community gathering places [16]. These patterns may also reflect gender-role differences, because women often have broader social networks and higher engagement in community activities.

This study has several limitations. First, residual confounding remains a concern owing to the non-randomized design of this quasi-experimental study. To mitigate this, we adjusted for baseline RSFD and 15 covariates, including demographic, socioeconomic, and health-related variables, using a mixed-effects linear regression model. This approach was chosen specifically to account for clustering by community, reflecting the fact that MEGURU STATION features varied by district policy and context. Additionally, the potential impact of unmeasured confounding was assessed using the E-value (CI limit = 2.01). Among covariates included in the model, only age and participation in neighborhood associations exceeded this threshold. Given that MEGURU STATION is often located near venues where neighborhood association meetings are held, the influence of unmeasured confounders beyond these factors is likely limited. While propensity-score methods (e.g., matching or weighting) were considered as an alternative strategy to further balance observed covariates, we did not apply them in this study due to concerns that our relatively small MEGURU STATION user sample (n = 187) could lead to unstable estimates and extreme weights. Future research with larger sample sizes should implement propensity-score approaches as an additional sensitivity analysis to strengthen causal inference. Second, this study did not identify whether the participants developed functional disability. Future studies using objective outcomes, such as the actual need for LTCI and mortality, will further strengthen the evidence on resource collection sites and functional disability. Third, although this study was conducted in multiple communities within specific regions of Japan, the generalizability of the findings to other cultural and geographic contexts may be limited. However, to address potential variability across communities, we applied a mixed-effects linear regression model, treating communities as random effects. This approach allowed us to account for differences between communities while assessing the overall association between MEGURU STATION use and the risk of functional disability. Nevertheless, unmeasured region-specific factors, such as local policies or social norms, may influence the results. Future studies should examine similar interventions in diverse cultural and

geographical contexts to explore broader applicability and context-specific adaptations. Specifically, verifying which model is more effective is desirable, considering factors such as walking distance to MEGURU STATION, frequency of use, sex differences, and combinations of peripheral functions. Additionally, the survey in Community A of Ikoma City coincided with the coronavirus disease pandemic, which may have affected social interaction and participation. Future studies should consider contextual factors to enhance generalizability. Finally, the baseline and follow-up response rates were 59.7% and 68.8%, respectively. Although these are typical for mail-based surveys, response bias may have affected the representativeness of the findings. Specifically, if non-respondents systematically differed from respondents—for example, if non-respondents were more socially isolated or had poorer health—the observed associations between MEGURU STATION use and RSFD could be underestimated.

Although MEGURU STATION was developed in a Japanese cultural context, its core concept—integrating opportunities for social participation into routine community activities—may be adaptable to various cultural and resource settings. In resource-constrained environments, similar models could be explored by incorporating social interaction into familiar community spaces, such as community markets or shared public areas, without requiring major infrastructure changes. Future implementation research across diverse contexts could help determine the feasibility and acceptability of adapting this model beyond Japan. In future studies, we plan to follow up with administrative data to evaluate actual LTCI certification and incidence of functional-disability incidence, thereby strengthening the evidence on the relationship between MEGURU STATION use and long-term care outcomes.

## Conclusions

This study contributes to a growing body of evidence demonstrating the importance of environmental interventions in maintaining health and preventing functional decline in older adults. By incorporating daily activities into preventive strategies, MEGURU STATION offers a possible method for reducing the risk of functional disability. Future studies should explore the long-term effects of these interventions and evaluate their applicability in different cultural and demographic contexts.

## Supporting information

**S1 Fig. Appearance of MEGURU STATION and its users.** (1) Ikoma City (Community A). (2) Tachiarai Town (Hongo community). (3) Tachiarai town (Ozeki community).
(DOCX)

**S1 Table. List of functions by MEGURU STATION installation area.**
(DOCX)

**S2 Table. Subjective changes due to MEGURU STATION use.** Note: Opportunities changed due to visits to MEGURU STATION installation sites in the past six months, from 1 year after installation. Data are presented as n (%). * Multiple answers were allowed.
(DOCX)

**S2 Fig. Residuals vs. Fitted Values plot.**
(DOCX)

## Acknowledgments

We thank the staff of the Ikoma City Community Care Promotion Division and the Tachiarai Town Welfare Division for their valuable contributions. We would like to express our gratitude to Amita Holdings Co., Ltd., for providing access to MEGURU STATION, which was instrumental in conducting this study. We also thank Editage (www.editage.com) for English language editing.

## Author contributions

**Conceptualization:** Noriyuki Abe, Kazushige Ide, Kenjiro Kawaguchi, Katsunori Kondo.

**Data curation:** Noriyuki Abe, Daisuke Kumazawa.

**Formal analysis:** Noriyuki Abe, Daisuke Kumazawa.

**Funding acquisition:** Kazushige Ide, Katsunori Kondo.

**Methodology:** Noriyuki Abe, Kazushige Ide, Kenjiro Kawaguchi, Daisuke Kumazawa, Katsunori Kondo.

**Project administration:** Katsunori Kondo.

**Supervision:** Kazushige Ide, Kenjiro Kawaguchi, Katsunori Kondo.

**Validation:** Noriyuki Abe.

**Visualization:** Noriyuki Abe.

**Writing – original draft:** Noriyuki Abe.

**Writing – review & editing:** Kazushige Ide, Kenjiro Kawaguchi, Daisuke Kumazawa, Katsunori Kondo.

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
