## [Decision Letter · Decision Letter 0]

30 Apr 2025

Dear Dr. Abe,

Thank you for submitting your manuscript to PLOS ONE. After careful consideration, we feel that it has merit but does not fully meet PLOS ONE’s publication criteria as it currently stands. Therefore, we invite you to submit a revised version of the manuscript that addresses the points raised during the review process.

We look forward to receiving your revised manuscript.

Kind regards,

Mario Ulises Pérez-Zepeda, M.D., Ph.D.

Academic Editor

PLOS ONE

 [1) Open Innovation Platform with Enterprises, Research Institute and Academia [OPERA, grant number JPMJOP1831] of Japan Science and Technology (JST).

2) Amita Holdings Co., Ltd.]. 

4. We note that Figure S1 includes an image of a participant in the study.

Reviewers' comments:

Reviewer's Responses to Questions

**Comments to the Author**

1. Is the manuscript technically sound, and do the data support the conclusions?

Reviewer #1: Yes

Reviewer #2: Partly

2. Has the statistical analysis been performed appropriately and rigorously?

Reviewer #1: Yes

Reviewer #2: No

3. Have the authors made all data underlying the findings in their manuscript fully available?

Reviewer #1: Yes

Reviewer #2: No

4. Is the manuscript presented in an intelligible fashion and written in standard English?

Reviewer #1: Yes

Reviewer #2: Yes

Reviewer #1: The manuscript presents a novel and well-executed quasi-experimental study that examines whether the use of a novel hub to reduce the risk of functional disability among older adults in Japan. The topic is timely and socially beneficial,aligning with current global priorities on healthy aging and social participation.

The concept is innovative to promote social participation. The quasi-experimental design is clearly compared between groups with stratified analyses. Results are presented with clarity with informative tables.

However, there are still some issues to be revised.The baseline and follow-up of one year require more exact timing of hub implementation using more detailed data collection. The perspective of using this Japanese community-based interventions for non-Japanese or lower-resource settings.The relative low response rate and follow-up rate could also be interpreted. Could the predictive tool be extended or updated for early functional disability screening, which could be mirrored by a more pronounced tendency in younger older adults and in women. PS. English grammar could be refined. Ethical approval and consent processes are clearly stated and conform to PLOS standards.

In brief, the study is of high relevance and technical quality with minor clarification and slight elaboration.

Reviewer #2: This quasi-experimental study evaluated whether older adults who use MEGURU STATION—a community hub built around resource collection—experience a lower risk of functional disability over one year compared with non-users. A total of 973 individuals aged ≥ 65 years from three Japanese communities completed baseline and 1-year follow-up surveys assessing the Risk Score for Functional Disability (RSFD). Mixed-effects linear regression (random effect = community), adjusted for demographic, socioeconomic, health, and social-participation covariates, found that MEGURU STATION use was associated with a modest but statistically significant reduction in RSFD (Coef. = –1.20; 95% CI: –2.27, –0.12; p = 0.029).

Sensitivity analyses using E-values and multiple imputation were conducted; additional self-reported measures demonstrated increased social interaction and community engagement among users.

General Comments

The manuscript addresses an important public-health question—how environmental interventions embedding social interaction into daily routines might delay functional decline. Strengths include:

A real-world, community-based intervention with high ecological validity.

Use of mixed-effects models to account for clustering by community.

Sensitivity analyses (multiple imputation, E-value) to probe robustness.

However, the quasi-experimental design and reliance on self-reported use impose limitations on causal inference and measurement validity. In addition, PLOS ONE policy requires full data availability, which is currently restricted by consent limitations.

Major Points

Causal Inference & Confounding

As a non-randomized study, residual confounding remains a concern. Although the E-value (CI limit = 2.01) suggests only strong unmeasured confounders could negate the association, baseline differences (users were older, more socially active, and had lower initial RSFD; Table 1)

warrant further adjustment or discussion. Propensity-score adjustment or matching could strengthen causal claims.

Clarify whether baseline RSFD was included as a covariate in the regression (methods mention adjustment for RSFD*). If not, adding baseline RSFD would control for regression to the mean.

Clinical Relevance of Effect Size

A coefficient of –1.20 on a 0–48 scale is statistically significant, but its practical impact on preventing long-term care certification is unclear. Please contextualize this change: e.g., what shift in absolute risk or number needed to treat does this represent?

Measurement of Exposure

MEGURU STATION “use” is self-reported at follow-up. Provide details on the survey question wording, frequency thresholds for classification as “user,” and any validation of self-report against logs or observations.

Data Availability

PLOS ONE requires that all data underlying findings be fully available. “Participants needed to provide consent for their data to be publicly shared” is insufficient. The authors should deposit de-identified data in a public repository or explain ethical/legal restrictions and provide a route for qualified researchers to access data.

Statistical Methods

Provide justification for choosing linear regression on RSFD rather than a generalized linear model if RSFD is skewed or bounded. Include diagnostic checks (e.g., residual plots).

Describe handling of missing data in covariates: percent missing, patterns, and convergence diagnostics for imputation.

Ethics & Funding Statements

Ethics approval and consent are clearly stated.

The funding statement lists two sources but lacks grant numbers associated with specific authors and funder URLs. Please format per PLOS guidelines.

Minor Points

Clarity and Style

In Table 1, clarify “Use of other MEGURU STATION functions*” – what counts as “other” versus core functions?

Replace “Coef.” with “B” (unstandardized beta) for consistency with Stata output.

Ensure uniform formatting of numbers (e.g., space before “%” in tables).

Introduction & References

Reference [19] (Amita HD marketing material) is a non–peer-reviewed source. Consider citing a peer-reviewed description or non-commercial report.

Check that all URLs in references are active and include access dates.

Figures & Supporting Information

Figure 1 (flowchart) is clear, but axes and labels (e.g., reasons for exclusion) could be enlarged for readability.

S1–S4 Tables are informative; consider moving S2 and S4 into the main text to highlight stratified and mechanism analyses.

Discussion

The discussion of cognitive stimulation via waste sorting is intriguing but speculative. Tone this down or cite direct evidence linking sorting tasks to cognitive benefit.

Recommendation

Revision required. Address major points—especially data availability, adjustment for baseline RSFD, and confounding—before reconsideration.

**Do you want your identity to be public for this peer review?** For information about this choice, including consent withdrawal, please see our Privacy Policy

Reviewer #1: No

Reviewer #2: No

---

## [Author Response · Author response to Decision Letter 1]

10 Jul 2025

We sincerely thank both reviewers for their detailed and constructive feedback. Below, we have provided point-by-point responses to each comment. All changes have been incorporated into the revised manuscript, with the tracked changes visible for clarity.

---

## [Editor Report · Decision Letter 1]

29 Aug 2025

Association Between Community-Based Resource Collection Site Use and Functional Disability Risk Among Older Adults: A Quasi-experimental Study

PONE-D-25-09359R1

Dear Dr. Abe,

We’re pleased to inform you that your manuscript has been judged scientifically suitable for publication and will be formally accepted for publication once it meets all outstanding technical requirements.

Kind regards,

Mario Ulises Pérez-Zepeda, M.D., Ph.D.

Academic Editor

PLOS ONE
---

## [Editor Report · Acceptance letter]

PONE-D-25-09359R1

PLOS ONE

Dear Dr. Abe,

I'm pleased to inform you that your manuscript has been deemed suitable for publication in PLOS ONE. Congratulations! Your manuscript is now being handed over to our production team.

Kind regards,

on behalf of

Dr. Mario Ulises Pérez-Zepeda

Academic Editor

PLOS ONE